# Study on the Method of Charge Accumulation Suppression of Electrostatic Suspended Accelerometer

**DOI:** 10.3390/s22134930

**Published:** 2022-06-29

**Authors:** Jiefeng Dai, Wenrui Wang, Bin Wu, Lingyun Ye, Kaichen Song

**Affiliations:** 1College of Biomedical Engineering and Instrument Science, Zhejiang University, Hangzhou 310027, China; 11515010@zju.edu.cn (J.D.); wubinzju@zju.edu.cn (B.W.); lyye@zju.edu.cn (L.Y.); 2School of Aeronautics and Astronautics, Zhejiang University, Hangzhou 310027, China; kcsong@zju.edu.cn

**Keywords:** charge accumulation, electrostatic suspended accelerometer, charge noise, combined waveform, acceleration noise suppression

## Abstract

Electrostatic suspended accelerometers (ESAs) are widely used in high accuracy acceleration measurement. However, there exist accumulated charges on the isolated mass which damage the accuracy and the stability of ESAs. In this paper, we propose to apply actuation voltage with a combined waveform to suppress the acceleration noise due to deposited charge. A model of the electrostatic force on the mass is established and the deviation voltage is found to be the dominant source of charge noise. Based on the analysis of disturbance electrostatic force under DC and AC signals, actuation combined with DC and AC voltage is designed and the disturbance force due to charge can be suppressed through adjustment towards the duty cycle of different compositions. Simulations and experiments are carried out and the results indicate that the disturbance due to charge can be suppressed up to 40%, which validates the efficiency of the scheme.

## 1. Introduction

Inertial sensors have become an indispensable part of navigation and geological survey systems thanks to the multiple conversion mechanisms and crafts [1]. Diversified mechanisms like resonant, piezo-resistive and piezoelectric mechanisms provide inertial sensors the capability to meet different measure ranges and bandwidths [2,3,4,5,6]. Among existing inertial sensors, electrostatic suspended accelerometers (ESAs) possess extremely high accuracy [7,8,9]. With the virtue of isolated sensitive body and high precision capacitance sensing, ESA is capable of micro acceleration measurement, especially high accuracy gravity measurement [10,11]. However, according to previous experiments, the sensitive body will acquire charges both in space operation and pre-test on the ground [12]. Due to the feature of non-contact, the accumulated charges on the mass cannot be neutralized directly [13,14,15,16]. As a result, the residual charge on the mass will change the equilibrium state of an ESA and introduce acceleration noise by coupling with nearby frames [17]. Consequently, the stability and the precision of the ESA are damaged.

Previous studies about accumulated charges in ESA focused on the factors from external environments and related facilities [13,14,15,16]. Several illustrations concerning the source of charge have been proposed as follows [14,18,19,20,21,22]: (1) Particles and cosmic rays in space environment, which lead to a charging rate up to 1.3 × 10^−11^ coulomb/day. (2) Triboelectric charging due to evacuation and large motion, which causes deposited charges of estimated 100 pC. (3) Emission electrons and UV radiation generated by venting facilities, which introduce deposited charges up to 3 × 10^11^ electrons. Based on the structure of ESAs, researchers also proposed schemes for suppressing deposited charges: (1) Discharge via gold wire. For satellites such as GOCE [19], CHAMP [20], and GRACE [21], this scheme is widely used. (2) Discharge by gas of high electron affinity [22]. (3) Neutralization by electron and ion beams [23]. (4) Discharge by UV radiation [24]. Discharge via electrical connection is the simplest method. However, the gold wire will introduce thermal noise and the circuit noise will couple with the test mass through the connection [25]. Contactless methods of neutralizing the deposited charge could cater for sensors with higher precision. However, there exist limitations due to the mechanism of the method. Schemes of adopting gas or particle beam can merely neutralize charges of single polarity since the beam only has affinity for the charge of single polarity. The UV method was applied in the GP-B mission and the LISA project for the feature of discharging without contact with the mass [8]. However, there still exist restrictions such as demand of charge measurement, coating damage and heating effect caused by high flux UV radiation, conflicting with the local electric fields in ESA [22,26,27,28]. In addition, the common restriction with existing contactless methods is the inability to perform charge neutralization while the sensor is in operation. Former studies supposed that the cosmic rays contribute most to the charge noise and existing schemes focused on charge transfer with the mass. In order to generate the desired electrification particles for neutralization, former contactless schemes require various auxiliary facilities for detecting and controlling the charge. Since the charge distribution on isolated mass depends on the distribution of voltage and the arrangement of electrodes, it is essential to take account of equivalent charge noise caused by a suspension system in practical terms, especially for ESA with a small distance between the mass and the electrodes [29,30,31]. In addition, such noise will be further aggravated by the demand of high suspension voltage for ground pretrial and large motion in orbit. Relevant research about charge noise derived from applied voltage has been rarely reported. At the same time, the electric field restricts the operation time and the discharge efficiency of existing schemes of charge management in ESA. Thus, there are still demands for methods to suppress charge noise on isolated sensitive mass.

In this paper, we propose a method for suppressing charge noise by applying time-varying actuation voltage. Based on the capacitor model of ESA, the analysis of electrostatic force with disturbance and acceleration noise due to different factors is represented. According to the characteristic of electrostatic force provided by DC and AC signals, a kind of actuation voltage with a combined waveform is proposed to suppress the charge noise and maintain the force efficiency. Referring to the parameters of ESA, the simulation model and testing structure were established and the comparison experiments were carried out to validate the suppression efficiency of the scheme. According to the result of the experiment, the time drift of electrostatic force due to deposited charge is suppressed up to 40% after adopting the scheme.

The rest of this paper is organized as follows. Section 2 is divided into three subsections. The model of electrostatic force in ESA is established in Section 2.1 and the factors that lead to charge noise are discussed as well. Section 2.2 proposes the method of suppressing charge noise via excitation while Section 2.3 illustrates the simulation results of the method. The descriptions of testing facilities and experiment results are presented in Section 3. Section 4 provides discussion about the effect of the scheme and brief comparison with other methods. Finally, the conclusions are presented in Section 5.

## 2. Theory

### 2.1. Model of Charge Distribution and Electrostatic Force in ESA

Aiming at enlarging the efficiency area for sensing and position control in ESAs, excitation voltage for detection and motivation control are both applied through fixed electrodes. Such an arrangement of electrodes has been applied in high precision systems like LISA [14]. In this work, the ESA we designed employs the same arrangement of electrodes. The parameter of the structure for analysis and the corresponding testing facility are exhibited in Figure 1.

ESA adopts the capacitance technique for displacement detection and equips electrodes with different functions to control the test mass. The actuation electrode on a single side is separated into differential pairs to maintain the virtual ground potential of the proof mass [31]. Based on the arrangement of the electrodes, the capacitor model of ESA is shown in Figure 2.

For ESA with closed-loop control, the proof mass is held in equilibrium position during operation and the electric field is constant. Assuming there is no charge transfer at first, the mass is in a condition of electrostatic equilibrium and remains electrically neutral. The distribution of charge in the steady electric field obeys the Gauss theorem [32]
(1)$$\nabla \cdot \vec{E}=\frac{\rho }{\varepsilon_{0}},$$
where *E* is the electric field on the surface, ρ is the surface charge density, $\varepsilon_{0}$ is the vacuum permittivity. The differential voltage $U_{j+}$ and $U_{j-}$ in single side can be simplified as the actuation voltage with constant amplitude. The charge distribution on the mass can be written as follows
(2)$$\Sigma \rho_{i1}S_{i1}+\Sigma \rho_{i2}S_{i2}+\Sigma \rho_{is}S_{is}+\Sigma \rho_{ie}S_{ie}=0,$$
where S is the effective projected area of the electrode on the mass. Considering of the external net charges deposited on the mass, the sensitive body will be in a new electrostatic equilibrium state and the potential of it will shift. Assuming that the quantity of the deposited charge under steady state is $Q_{e}$, Equation (2) can be further changed into
(3)$$\left(U_{1}-U_{e}\right)\cdot \Sigma C_{1}+\left(U_{2}-U_{e}\right)\cdot \Sigma C_{2}+\left(U_{s}-U_{e}\right)\cdot \Sigma C_{s}+\left(U_{g}-U_{e}\right)\cdot \Sigma C_{e}+Q_{e}=0,$$
where $U_{e}$ is the potential of the proof mass, $U_{g}$ is the earth potential and generally zero. The equivalent capacitance of ESA can be seen as constant when the mass is held in equilibrium position. The potential of the mass can be described as a combination of net charge and the applied voltage from surrounding electrodes
(4)$$U_{e}=U_{i}+\frac{Q_{e}}{C_{TM}}$$
where
(5)$$C_{TM}=\Sigma C_{j}+\Sigma C_{s}+\Sigma C_{e},$$
(6)$$U_{i}=\frac{\Sigma C_{j}\cdot U_{j}+\Sigma C_{s}\cdot U_{s}}{C_{TM}},$$

For ESA with strictly symmetric voltage distribution and geometrical structure, the induced voltage $U_{i}$ is zero while in equilibrium position. Ignoring the coupling effect between electrodes with different functions, the electrostatic force on the mass in single direction can be expressed as [18,33]
(7)$$F_{e}=-\frac{1}{2}\left[\Sigma \frac{\partial C_{j}}{\partial k}{\left(U_{j}-U_{e}\right)}^{2}+\Sigma \frac{\partial C_{s}}{\partial k}{\left(U_{s}-U_{e}\right)}^{2}+\Sigma \frac{\partial C_{e}}{\partial k}U_{e}^{2}\right],$$
where k is the generalized coordinate for the sensitive axis. Substituting Equations (5) and (6) in (7), the formula can be transformed into [18,33]
(8)$$F_{e}=\frac{1}{2}\cdot \frac{\partial C_{j}}{\partial k}\cdot \Sigma {\left(U_{j}-U_{i}\right)}^{2}+\frac{1}{2}\cdot \frac{\partial C_{s}}{\partial k}\cdot \Sigma {\left(U_{s}-U_{i}\right)}^{2}+\frac{Q_{e}^{2}}{2C_{TM}^{2}}\cdot \frac{\partial C_{TM}}{\partial k}-Q_{e}\frac{\partial }{\partial k}\left(U_{i}\right),$$

The first term provides the control authority for position adjustment. The second term is the force introduced by the sensing voltage. The last two terms indicate the electrostatic force associated with deposited charge on the mass. The third term shows the relationship between the capacitance gradient and deposited charge on the mass. The last term is the interaction between induced voltage and deposited charge on the mass. All the other terms except the first term are disturbance forces that will be reflected in the measurement of acceleration. Although there exist other disturbances introduced by the control strategy or ambient temperature, the effects can be mitigated by algorithmic optimization or temperature compensation and the disturbance force due to deposited charge cannot be ignored [34,35,36]. According to Equation (8), the disturbance force is related to the quantity of deposited charge, the applied voltage and the position deviation. Thus, the corresponding acceleration noise due to possible influence factors can be described as [23]
(9)$$S_{a_{e}}^{1/2}={\left(S_{Q_{e}}{\left(\frac{\partial a_{e}}{\partial Q_{e}}\right)}^{2}+S_{U_{j}}{\left(\frac{\partial a_{e}}{\partial U_{j}}\right)}^{2}+S_{k}{\left(\frac{\partial a_{e}}{\partial k}\right)}^{2}\right)}^{\frac{1}{2}},$$

The acceleration $a_{e}$ can be attained as $a_{e}=F_{e}/m$, where m is the mass of the sensitive body. Generally, the sensing voltage is applied in sinusoidal form with high frequency and the amplitude of it is small. As a result, the disturbance caused by it can be ignored and the charge-dependent noise contributes most to the total disturbance. In other words, the essential purpose of suppressing charge accumulation is to reduce the influence of deposited charge. Among the listed factors, existing schemes focus on reducing the quantity of net charge in the milli-Hz region and the disturbance introduced by actuation voltage is assumed as white noise. Furthermore, the source of deposited charge was ascribed to cosmic rays as a result of the operating environment.

However, the actuation voltage contains fluctuation noise due to the restriction of output facilities in practical terms. Meanwhile, there exist certain factors that cause stray electric DC fields as thermal noise and patch effect. According to the experiment, the noise from the high-voltage amplifying circuit is on the order of 2.0 × 10^−4^ $V/\sqrt{\mathrm{Hz}}$ at 0.1 Hz while providing 766 V for suspension, which is greater than estimated [37]. Therefore, the distribution of applied voltage is asymmetric, especially for applications with broader bandwidth. At the same time, the potential drift of the isolated mass can be attributed to the cooperative effect of net charge and asymmetric voltage distribution. Based on the characteristic of isolated mass, we supposed the absolute deviation voltage to be an additional source superimposed on the actuation source and assigned it to single side of the differential electrode. We define the absolute deviation as $U_{x}$ and the sign of $U_{x}$ is contained in the variable. Thus the applied voltage with deviation can be described as
(10)$$U_{jx}=U_{j}+U_{x},$$

The potential of the mass can be transferred into
(11)$$U_{ex}=U_{ix}+\frac{Q_{e}}{C_{TM}}=U_{i}+\frac{Q_{e}+\Sigma C_{j}\cdot U_{x}}{C_{TM}}=U_{i}+\frac{Q_{ex}}{C_{TM}},$$

Substituting Equations (10) and (11) in (8), the electrostatic force with asymmetric voltage distribution can be attained as
(12)$$F_{ex}=\frac{1}{2}\cdot \frac{\partial C_{j}}{\partial k}\cdot \Sigma {\left(U_{jx}-U_{i}\right)}^{2}+\frac{Q_{ex}^{2}}{2C_{TM}^{2}}\cdot \frac{\partial C_{TM}}{\partial k}-\frac{Q_{ex}}{C_{TM}}\cdot \frac{\partial }{\partial k}\left(\Sigma C_{j}\cdot U_{jx}\right),$$

According to Equation (12), the deviation voltage leads to additional deposited charges on the mass, which will be reflected in the disturbance force. Although the efficient force also changes, the last two terms are still the dominant disturbance. Taking the parameter of the prototype ESA we designed as reference, the total capacitance is 2.066 nF and the actuation capacitance in single direction is 0.5165 nF. The mass of the sensitive body is estimated as 0.054 kg, the distance between the sensitive body and the electrode is 0.3 mm. The contributions of influence factors towards the acceleration noise are shown in Figure 3.

In accordance with Figure 3, it can be seen that the noise caused by voltage fluctuation is greater than that due to charge. Furthermore, the induced charge introduced by deviation also contributes to the charge noise. According to the calculation, the deviation voltage of 1 mV results in charge noise equal to the one due to net charge of 18.33 pC and the quantity of induced charge will increase with the rise of the capacitance. For the ESA in pre-test on the ground, the voltage fluctuation is greater and the incident charge is blocked by housing and atmosphere. Therefore, it is essential to take account of the charge noise due to deviation voltage and it will be the dominant noise source in certain cases.

### 2.2. Method of Suppressing Charge Accumulation

#### 2.2.1. Charge Noise under AC Signals

The acceleration noise introduced by deviation voltage cannot be ignored and will aggravate with DC voltage drift. Meanwhile, the quantity of external charge can be approximated as a linear function of time [18]. Since the distribution of charge on isolated mass depends on the applied voltage, we proposed to apply actuation voltage which is variable in the time domain to suppress the disturbance due to residual charge. To start with, it is essential to model the electrostatic force and charge noise under time-varying actuation. In practical terms, the test mass in ESA is treated with a conductive coating and the induced charge will assemble on the surface of the mass rather than being restrained. Therefore, the distribution of charge density in the time domain also follows the Gauss theorem. We choose a signal in sinusoidal form due to the time continuity and lesser harmony. The absolute deviation is reflected in the amplitude of actuation voltage due to the modulation effect of AC signals. Meanwhile, since the deviation voltage is the cooperation of different factors, the amplitude of it is consistent with the one under DC actuation. Assuming there is no frequency difference, the disturbance signal can be attained as a sinusoidal signal with the frequency being the same as actuation voltage. Based on Equation (12), the electrostatic force with disturbance term when applying actuation in sinusoidal form can be described as
(13)$$F_{ex}\left(t\right)=\frac{1}{2}\cdot \frac{\partial C_{j}}{\partial k}\cdot \Sigma {\left(U_{jx}\left(t\right)-U_{i}\left(t\right)\right)}^{2}+\frac{Q_{ex}^{}{\left(t\right)}^{2}}{2C_{TM}^{2}}\cdot \frac{\partial C_{TM}}{\partial k}-\frac{Q_{ex}\left(t\right)}{C_{TM}}\cdot \frac{\partial }{\partial k}\left(\Sigma C_{j}\cdot U_{jx}\left(t\right)\right),$$
where
(14)$$U_{i}\left(t\right)=\frac{\Sigma C_{j}\cdot U_{j}\left(t\right)+\Sigma C_{s}\cdot U_{s}\left(t\right)}{C_{TM}},$$
(15)$$U_{jx}\left(t\right)=(U_{j}+U_{x})\sin\left(2\pi f_{a}t\right),$$
(16)$$U_{j}\left(t\right)=U_{j}\sin\left(2\pi f_{a}t+\varphi_{j}\right),$$
(17)$$U_{s}\left(t\right)=U_{s}\sin\left(2\pi f_{s}t+\varphi_{s}\right),$$
(18)$$Q_{ex}\left(t\right)=\lambda_{e}\cdot t+\Sigma C_{j}\cdot U_{x}\sin\left(2\pi f_{a}t\right),$$
where $\lambda_{e}$ is the average charging rate, $f_{a}$ is the frequency of the actuation voltage, $f_{s}$ is the frequency of the sensing voltage, $\varphi_{j}$ and $\varphi_{s}$ are the phase offsets. The frequency of applied voltage is far from the measurement bandwidth of ESA. Substituting Equations (13)–(18), the disturbance force due to deposited charge under AC actuation can be described as
(19)$$\begin{array}{l} F_{eQ}\left(t\right)=\frac{{\left(\lambda_{e}\cdot t+\Sigma C_{j}\cdot U_{x}\sin\left(2\pi f_{a}t\right)\right)}^{2}}{2C_{TM}^{2}}\cdot \frac{\partial C_{TM}}{\partial k}\\ -\frac{\left(\lambda_{e}\cdot t+\Sigma C_{j}\cdot U_{x}\sin\left(2\pi f_{a}t\right)\right)}{C_{TM}}\cdot \frac{\partial }{\partial k}\left(\Sigma C_{j}\cdot U_{j}\left(t\right)\right) \end{array}$$

The equation above can be further simplified as
(20)$$F_{eQ}\left(t\right)≅k_{1}\cdot \lambda_{e}^{2}\cdot t^{2}+k_{2}\cdot {\left(U_{x}\cdot C_{j}\right)}^{2}\left(1-\cos\left(4\pi f_{a}t\right)\right)+k_{3}\cdot \lambda_{e}\cdot t\cdot C_{j}\cdot U_{x}\sin\left(4\pi f_{a}t\right),$$

Similarly, the disturbance force under DC actuation can be attained as
(21)$$F_{edcQ}\left(t\right)≅k_{1}\cdot \lambda_{e}^{2}\cdot t^{2}+2\cdot k_{2}\cdot {\left(U_{x}\cdot C_{j}\right)}^{2}+k_{3}\cdot \lambda_{e}\cdot t\cdot U_{x}\cdot C_{j},$$
where $k_{1}$, $k_{2}$ and $k_{3}$ are constants related to the structural features of the ESA. The first term is the disturbance force due to external charge and is exponential to $t^{2}$. The second term indicates the disturbance force due to induced charge. The third term represents the correlation between net charge and deviation voltage. Since the frequency $f_{a}$ is far from the bandwidth of ESA, the disturbance force related to $f_{a}$ will not be reflected in the measured acceleration. Supposing the absolute deviation is also time-varying, the disturbance force correlated to time can be further transferred into
(22)$$\begin{array}{l} F_{eQc}\left(t\right)≅k_{1}\cdot \lambda_{e}^{2}\cdot t^{2}+k_{2}\cdot {\left(U_{x}\left(t\right)\cdot C_{j}\right)}^{2}\left(1-\cos\left(4\pi f_{a}t\right)\right)\\ +k_{3}\cdot \lambda_{e}\cdot t\cdot C_{j}\cdot U_{x}\left(t\right)\sin\left(4\pi f_{a}t\right) \end{array}$$
where $\lambda_{x}$ is the approximate drift coefficient of deviation voltage and the disturbance force is suppressed as well. The prototype ESA we designed possesses multi-electrode pairs to attain higher electrostatic force. Aiming at simplifying the calculation, we chose one set of electrodes for analysis and the parameters are shown in Table 1. The deviation of voltage is set as constant and the charging rate is expected to be +1100 e/s according to observation [5,25]. The amplitude of AC signals is higher to cover the efficiency loss. The disturbance force and the total electrostatic force under different forms of actuation are shown in Figure 4.

According to the calculation, the peak value of disturbance force while applying AC actuation is approximate to the one under DC actuation and the frequency of sinusoidal varying force is 1000 Hz. Under given conditions, the disturbance force is mainly contributed by deviation voltage and the bias of AC signal increases with the rise of net charge. Through the introduction of AC signals, the disturbance force due to deposited charge noise can be suppressed up to 50%.

#### 2.2.2. Scheme of Suppressing Charge Noise by Actuation with Combined Waveform

Based on the calculation above, AC actuation is confirmed to be efficient for suppressing charge noise originating from deviation voltage. However, there exist several drawbacks of merely adopting AC actuation. Compared with DC actuation, the amplitude for suspension needs to be higher, which leads to the tendency of electrical breakdown and the increase of emission current.

Based on the features of DC and AC actuation, we propose a kind of actuation voltage with a combined waveform which is periodic in time domain. Focused on the requirement of long-term operation, the applied voltage is required to suppress the disturbance due to deposited charge and offer suspension force for high input. Thus, the waveform we designed is divided into two parts in a single period. The first part is the DC signal for force efficiency and the second part is the sinusoidal signal for charge dissipation acceleration [38]. According to Equation (7), the electrostatic force is proportional to the square of the applied voltage. Thus, for the time-varying signal, the electrostatic force offered by it can be attained through the integral with respect to time. We define a variable n to describe the duty cycle of the signal. For the control system of the ESA shown in Figure 5, the coefficient n is a variable acting on $H_{a}$ and leads to the variation of feedback voltage for balancing the input force.

While adopting actuation in the combined waveform, the coefficient of efficient force $\eta_{e}$ can be described as:(23)$$\eta_{e}=\frac{1}{TU_{j}^{2}}\left[{\int_{T/n}^{T}\left(U_{j}\sin\left(2\pi f_{a}t\right)\right)}^{2}dt+\int_{0}^{T/n}{\left(U_{j}\right)}^{2}dt\right],$$
where *T* is the single period of the signal and $U_{j}$ is the amplitude of the actuation voltage. In order to suppress the additional disturbance, the frequency of signal switching must be higher than the bandwidth of the target signal. The combined waveform is shown in Figure 6.

The coefficient $\eta_{e}$ is 1 when the electrostatic force is provided only by DC voltage and the amplitude of the actuation voltage must be higher when $\eta_{e}<1$. We use a normalization coefficient $\eta_{a}$ to describe the magnification ratio, which can be attained as $\eta_{a}\propto 1/\sqrt{\eta_{e}}$. The relationship between the two coefficients and the duty cycle is shown in Figure 7.

In accordance with Figure 7, the efficient electrostatic force decreases proportionately to the reciprocal of n and the amplitude of necessary suspension voltage increases with the enlarged constituent of the AC signal. Parallel to the efficient electrostatic force, the disturbance force under actuation with the combined waveform can be described as:(24)$$\begin{array}{l} F_{eQT}\left(t\right)≅k_{1}\cdot \lambda_{e}^{2}\cdot t^{2}+k_{2}\cdot {\left(U_{x}\left(t\right)\cdot C_{j}\right)}^{2}\cdot \left(1-\cos(4f_{a}\pi t)+\cos(\frac{4f_{a}\pi t}{n})\right)\\ +k_{3}\cdot \lambda_{e}\cdot t\cdot C_{j}\cdot \frac{1}{n}\cdot U_{x}\left(t\right)\left(1+n\sin(2f_{a}\pi t)-\sin(\frac{2f_{a}\pi t}{n})\right) \end{array}$$

Taking the parameters in Table 1 as reference, the changes of total electrostatic forces under different forms of actuation and different duty cycles are shown in Figure 8.

Based on the given condition, the disturbance force is suppressed by 40% under pure AC actuation and 20% under combined waveform. According to Equations (20)–(22), it can be seen that the drift of electrostatic force is proportional to the deposited charge. Thus the variation of the force can be utilized to estimate the charge accumulation. In order to describe the suppression effect of the method directly, we used a normalization coefficient as “suppression efficiency” to indicate the decline of charge noise. In this work, the suppression efficiency is defined as the ratio of force drift under DC actuation to the one under the same conditions except for the configuration of applying voltage.

Although the suppression efficiency of the scheme is lower, it can be seen that the time drift of electrostatic force decreases with the rise of coefficient n and the suppression efficiency can be promoted by extending the composition of the sinusoidal signal in a single period. The variation of suppression efficiency is shown in Figure 9.

In accordance with Figure 6 and Figure 8, the amplitude of suspension voltage and suppression efficiency is adjustable through the variation of coefficient n. For occasions that require a high dynamic range, the coefficient n is set approximately to 1 to provide force efficiently. For occasions that require long stability and high precision, the coefficient n is increased to promote the charge noise suppression efficiency. In order to verify the suppression effect of the waveform with the combined waveform, we simulated the response with the noise signal based on the control model shown in Figure 4. In accordance with the simulation, the errors of acceleration measurement under DC actuation and combined actuation are shown in Figure 10.

In accordance with Figure 10, the amplitude of acceleration error can be suppressed up to 40% while n increases. To sum up, the capability of noise suppression and the measurement range can be maintained at the same time while adopting the scheme. In order to verify the validity, we further analyzed the scheme through the following simulation experiment.

### 2.3. Simulation

Based on the capacitor model of ESA and structural parameters shown in Table 1, a three-dimensional model was established with COMSOL Multiphysics software to analyze the electrostatic force and the distribution of the charge. In order to respond to the theoretical calculation, we also chose one set of the electrodes for analysis and the diameter of the electrode was same as in reality. The model is combined with two sets of differential electrodes and a pair of sensing electrodes, which are all set as polished metals. In practical terms, the system is closed-loop. Thus the proof mass was fixed at the equilibrium position and the initial charge on the mass was zero. All of the electrodes and the mass follow charge conservation and the permittivity of the medium was set as 1 (same as the atmosphere). The actuation voltage was configured symmetrically on one side of the model and other electrodes were grounded to avoid unwanted disturbance. In accordance with the simulation, the distributions of surface charge and electric field under DC actuation are shown in Figure 11.

In accordance with Figure 11, charges with different polarities assemble on the projection plane of actuation electrodes. Based on the electrode arrangement of ESA, we define the surface which adjoins grounded electrodes as the neutral region. In accordance with Equations (1) and (3), the surface charge density in the neutral region indicates the potential deviation. The simulation calculation also exhibits that the charge density in all neutral region remains same. Under given boundary conditions, the surface charge density of the neutral region when voltage fluctuation occurs is displayed in Figure 12. The voltage Va and Van indicate the actuation applied on the differential electrodes.

In accordance with Figure 11, the charge density in neutral area arises with the deviation of the actuation voltage and the absolute deviation of 1 mV leads to an increment charge of 5.58 fC on the mass. The spatial distribution of charge at discrete time points while applying AC actuation voltage is shown in Figure 13.

In Figure 12 and Figure 13, it can be seen that the spatial distribution of charge is independent of the actuation waveform. Therefore, the surface charge density in the neutral region is a basis of potential drift regardless of the actuation waveform. The electrostatic force and surface charge density in the neutral domain under different forms of actuation while the deposited charge on the mass is linearly increasing in the time domain are shown in Figure 14.

In accordance with Figure 14, the drift of electrostatic force is 1.838 × 10^−10^ N under DC actuation and 1.247 × 10^−10^ N under AC actuation. The disturbance force due to the deposited charge is suppressed by 32 % and the surface charge density under AC actuation exhibits the feature of sinusoidal signal. Taking account of the deviation voltage, the electrostatic force and the surface charge density of the mass under different forms of actuation are shown in Figure 15.

In accordance with Figure 15, the drift of electrostatic force is 1.838 × 10^−10^ N under DC actuation and 1.222 × 10^−10^ N under AC actuation. The disturbance force due to the deposited charge is suppressed by 34% and there exists a constant offset of electrostatic force due to the deviation voltage. In accordance with the simulation, it can be seen that the disturbance force due to the deviation voltage is higher than the one due to the net charge. Meanwhile, the surface charge density tends to be symmetric to zero arising with the deviation voltage under AC actuation. The electrostatic force and surface charge density while applying actuation in the combined waveform are shown in Figure 16.

In accordance with Figure 16, the efficient electrostatic force is 4.67821 × 10^−6^ N under combined waveform actuation and it is 73.47% of the force provided by DC actuation with the same amplitude. The efficient electrostatic force and surface charge density on different occasions while the charge is increasing linearly are shown in Figure 17.

In accordance with Figure 17, the drift of electrostatic force is 1.511 × 10^−10^ N for symmetric actuation and 1.451 × 10^−10^ N for asymmetric actuation. The suppression efficiency increased from 18% to 21% while taking account of the deviation voltage. In accordance with the simulation, the suppression efficiency also increases with the expansion of the composition of the sinusoidal signal. The suppression efficiency under different forms of actuation is shown in Table 2.

In the simulation model, the adjusted factor was restricted to the net charge and the deviation voltage. Thus, the drift of electrostatic force can be attributed to the effect of the deposited charge, which is combined with net charge and induced charge. The suppression efficiency exhibited in Table 2 indicates that the disturbance force is suppressed and the suppression effect is improved while the deviation voltage increases.

In accordance with the simulation result, the magnitude of electrostatic force is approximate to the theoretical calculation and the force drift is a long-term variation in the time domain. Unfortunately, although the scheme is proven to be efficient, the suppression efficiency calculated with the simulation is lower than the theoretical result with the same parameters. In accordance with the electrostatic force model shown in Equation (13), the total disturbance force is the subtraction of the last two terms and the numerical solution of it is related to the polarity of the charge and the deviation voltage. Due to the grid limitation of the three-dimensional model, there exists an initial offset of the mass potential; even the actuation is configured symmetrically and the offset increases while applying AC actuation. Based on Equation (22), the increase of potential offset aggravates the disturbance force. We reduced the offset as much as possible by refining the mesh and adding ground boundaries. Although the simulation result exhibits that the initial charge density is approximately zero, the offset cannot be dismissed drastically. In addition, the distortion while processing continuous actuation signals causes error as well. Therefore, the disturbance force under AC actuation is amplified and the suppression efficiency is reduced. In accordance with the measurements of existing studies, the potential offset will not be intensified while applying AC actuation in practical term [39]. This conclusion provides the basis for us to configure symmetrical actuation in the experiment as expected.

For a closed-loop system, the variation of electrostatic force is reflected by feedback voltage and the drift will be exhibited as displacement offset. Based on calculation, the theoretical feedback voltage under given parameters is approximately 40 μV to balance the force drift, which is difficult to supply. Furthermore, the equivalent electrostatic stiffness and the noise of other sub-systems will introduce unexpected disturbance. As a result, the drift of electrostatic force cannot be separated from other factors. In accordance with the curve of simulation, the electrostatic force decreases linearly with time and the slewing rates under the same actuation approximate each other. Therefore, we introduced a time-invariant force into the experiment and established an equilibrium relationship between two kinds of force. Furthermore, the time-invariant force can be utilized as a basis to obtain the electrostatic force indirectly.

## 3. Experiments

In accordance with the analysis, the deposited charge leads to a time drift of electrostatic force. In order to obtain the electrostatic force in long-term operation, a simplified suspension pendulum was established as shown in Figure 18. The suspension pendulum is appropriate for measuring weak force due to its low stiffness and has been used in micro-vibration measurement systems [5]. Since the electrostatic force is weak, the force application axis is set in the horizontal direction to utilize the advantage of the pendulum. In response to the theoretical calculation, the arrangement of the electrodes and the configuration of the applied voltage are the same as the simulation model. We regarded gravity as an invariant force and arranged it in the orthogonal direction with the electrostatic force. While applying actuation, the deflection angle will occur and can be used to calculate the electrostatic force. In order to restrict unexpected disturbance, the system was built as an open-loop system and the displacement of the mass in the horizontal direction indicates the electrostatic force.

The sensitive body in the vertical structure is a 30 mm × 1.1 mm gold-coated wafer made from aluminum oxide ceramics and the mass of the wafer is 11.8 g. Aiming at limiting the move direction to horizontal, the wafer is suspended by two Kevlar fibers 26.35 cm long. The high tensile strength of the fiber ensures that the length will not change during the experiment. The efficient area of the actuation electrodes is 9.176 × 10^−4^ m^2^ and is divided into two parts to raise differential actuation. In accordance with the simulation calculation, the test mass will move within a small range of 15 μm near the equilibrium position while introducing electrostatic force. The variation of displacement is far less than the initial gap and the motion process can be linearized. Considering that the force drift varies slowly, the measurement is based on the stationary state of the mass. In order to dismiss the air damping between electrodes, we increased the gap between the mass and the electrodes to 1.4 mm and the capacitance of the actuation electrode in single side is 5.41 pF. The amplitude of the actuation was also increased to compensate for the loss of the electrostatic force. The displacement of the wafer was measured with a laser displacement sensor OPTEX CD33-120NV, which was connected to the computer via NI cDAQ-9171. The amplitude of actuation voltage was measured with an oscilloscope TBS 1102 and kept constant while in operation. The testing structure was placed in a shielding chamber to avoid the disturbance due to temperature and air flow.

The displacement variation of the suspended wafer in the time domain while applying actuation in different forms on the electrode is shown in Figure 19.

From Figure 19, the time drift of displacement under DC actuation is approximately 0.00455 mm/h, the one under AC actuation is approximately 0.00085 mm/h and the one under combined waveform is 0.003 mm/h. The drift of displacement can be transferred into the variation of electrostatic force through the corresponding relationship with gravity. Since the actuation voltage was observed and kept symmetric during the measurement, the force drift is attributed to the deposited charge from the atmosphere and stray electric field on the mass. Based on the equation represented before, we convert the drift of electrostatic force into the equivalent charge on the mass. Hence, the equivalent deposited charge is 13.015 pC under DC actuation, 5.625 pC under AC actuation and 10.568 pC under combined waveform. The disturbance due to charge is suppressed by 56.8% while applying AC actuation. The suppression efficiency is higher than the calculation due to the blocking effect of the AC electric field on the emitted charge. The displacement in the time domain under the combined waveform while the coefficient n is increasing is shown in Figure 20.

In accordance with Figure 20, the electrostatic force decreased with the increase of n and the attenuation ratio of the force is approximately 1/n. The time drift of displacement and the equivalent deposited charge are shown in Table 3.

In accordance with Table 3, the suppression efficiency increases with the coefficient and it varies obviously while n is close to 2, which is consistent with the results of the theoretical derivation.

In order to verify the process of charge dissipation under different actuation, we also measured the variation of displacement after switching the source as shown in Figure 21. In accordance with the figure, the recovery time is approximately 0.9 s under the combined waveform and 7.4 s under DC actuation. Thus, the scheme of applying actuation in the combined waveform is also validated for accelerating the charge dissipation.

## 4. Discussion

In this study, we explored the constitution of charge noise and discussed the method of suppressing charge noise based on the electrostatic force model. The simulation and the experiment results imply that the force drift due to deposited charge is suppressed via the actuation with combined waveform. Likewise, AC actuation is identified to be effective for depressing charge noise. In accordance with the electrostatic force model, AC actuation affects the disturbance from the interaction between deposited charge and deviation voltage; meanwhile, the noise due to net charge will not be suppressed. The experiment results provide a basis for the inference that the deviation voltage is a dominant source of charge noise. Consistent with the existing method of stray field compensation, the potential drift due to deposited charge can be compensated for by applying certain excitation [29,39]. The difference is that our study focused on the continuous disturbance in time domain and the proposed scheme is identified to be effective for suppressing the time-dependent charge noise.

Existing methods of charge management neutralize the deposited charge by active charge transfer. In accordance with the scheme of connecting the mass, these methods can be divided into direct schemes and indirect schemes. A direct scheme like gold wire could diminish the deposited charge immediately and maintain the potential of the mass [19,25]. However, the structural thermal noise and the electrical noise introduced by wire depress the precision of the sensor. Especially for the application with high suspension voltage, the noise contained in high voltage will couple with the mass through the wire. Therefore, the contactless scheme is more conductive to a high-precision system. Our scheme is also proposed for this reason and adopts an indirect scheme. For indirect schemes like gas, ion beam and UV radiation, the restriction of the schemes can be summarized as three points [23,25,26,40]: (1) The discharging process needs to be employed in the interval of sensor operation. (2) The requirement of complex auxiliary equipment. (3) The discharging rate and the polarity of neutralized charge are limited. Since the motion of charged particles is affected by the electric field, it is required that the actuation voltage should be switched off while adopting the existing scheme. Therefore, a budget of deposited charge is required to determine when to start charge neutralization. Compared with existing methods, our scheme focuses on suppressing the disturbance due to deposited charge rather than neutralizing the charge on the mass. The actuation with combined waveform can provide necessary electrostatic force to feedback input force and the sensor is still in operation. Since the disturbance is suppressed by actuation voltage, the additional auxiliary mechanism is not required, which reduces the burden on the structure and control system. Although the net charge cannot be eliminated, the operation time of the ESA is extended and the proposed scheme can be considered complementary to existing methods for raising the budget. Moreover, it has been verified that in some cases the deviation voltage is the dominant noise source, which can be eliminated by the proposed scheme.

Despite its preliminary character, this study can clearly indicate the suppression effect of the scheme proposed. The facility we used for measurement is a principal verification structure and there exists difference in the detailed configuration compared with mature sensors. For example, the actuation is supplied by a transformer for the necessary amplitude of the voltage. The precision of the signal is lower than the one provided by high resolution DAC, which increases the disturbance due to deviation. The efficiency loss while switching the waveform also leads to a reduction in electrostatic force. Therefore, the noise performance of the proposed method in mature ESAs is worthy of study. Moreover, the control strategy of adjusting the actuation must be explored while adopting the scheme in a practical system. The precision of actuation voltage must be promoted in future research as well.

## 5. Conclusions

In this paper, we proposed a method of applying actuation with combined waveform to suppress the disturbance due to deposited charge. The results of the simulation and the experiment indicate that the disturbance force is suppressed by introducing AC actuation. Based on the equivalent model of ESA, the electrostatic force of the mass was analyzed to identify the source of charge noise. It was found that deviation voltage leads to additional charge noise, which becomes a dominant disturbance on certain occasions. Based on the characteristic of actuation with different forms, we designed a kind of actuation with combined waveform to relieve the conflict between the suppression efficiency and the force coefficient. The experiment on the suspension pendulum shows that the time drift of the electrostatic force can be suppressed up to 40% under the designed actuation and the force coefficient of the actuation with same amplitude can be adjusted by duty cycle variation. The results of simulation and experiment indicate that our scheme is beneficial for promoting the precision and the operation time of ESA without complicated auxiliaries. Future work will focus on optimizing the control strategy to adapt different inputs and promoting the stability and precision of the actuation voltage.

## Figures and Tables

**Figure 1 sensors-22-04930-f001:**
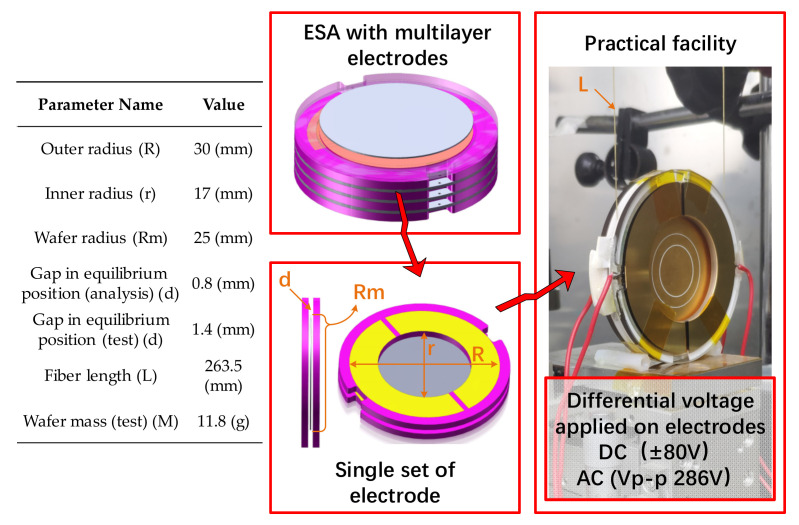
Structure parameters and the overview of the device for analysis.

**Figure 2 sensors-22-04930-f002:**
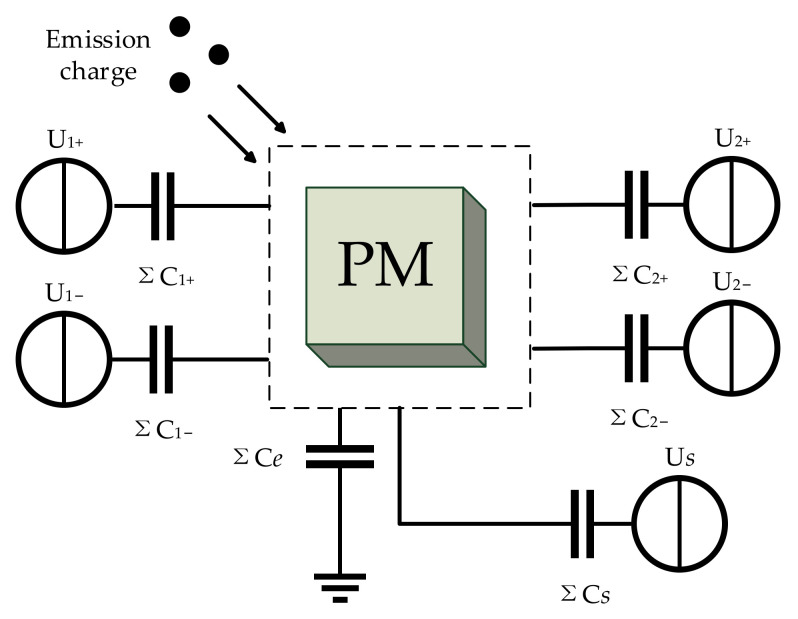
Capacitor model of ESA in single axis. Capacitances with the same subscript number are equal to each other. Where $U_{j}$ is the actuation voltage applied on the electrode, $C_{j}$ is the actuation capacitance, $U_{s}$ is the excitation voltage for sensing, $C_{s}$ is the sensing capacitance, $C_{e}$ is the coefficient of capacitances between the mass and the grounding frames.

**Figure 3 sensors-22-04930-f003:**
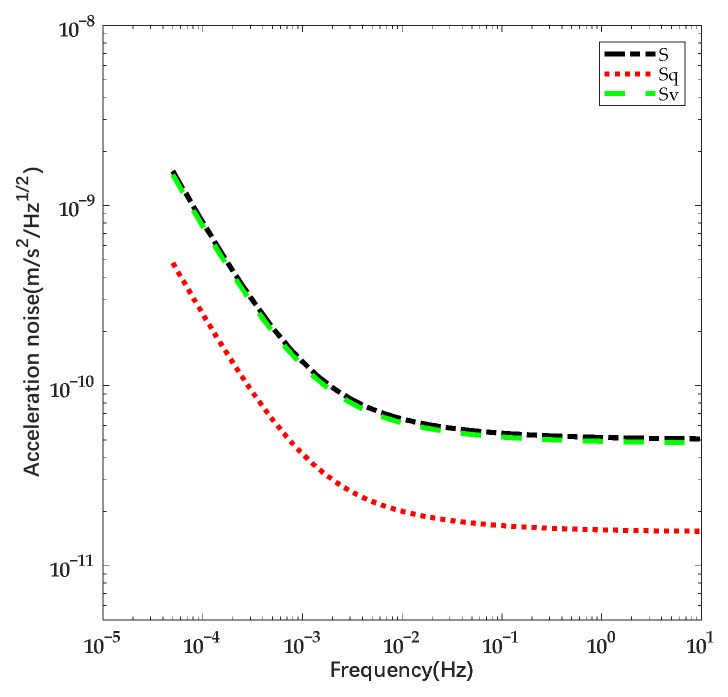
Estimation of acceleration noise caused by listed factors, S presents the total noise, $S_{q}$ presents the noise due to emitted charge and $S_{v}$ presents the noise due to deviation voltage.

**Figure 4 sensors-22-04930-f004:**
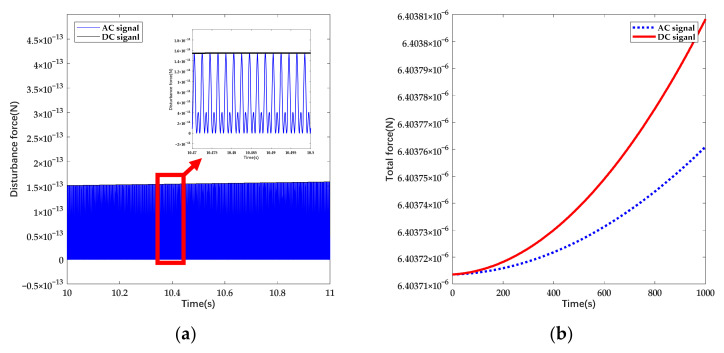
The variation of disturbance force and total force with linearly increasing deposited charge in time domain: (**a**) Original calculation result of disturbance force; (**b**) Total electrostatic force; the force under AC actuation is attained as effective value.

**Figure 5 sensors-22-04930-f005:**
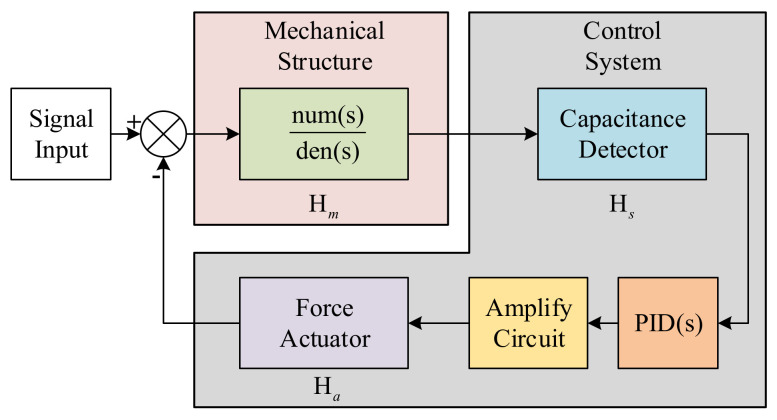
The schematic model of control system of ESA. $H_{m}$ denotes the transfer function of mechanical structure, $H_{s}$ denotes the transfer function of capacitive displacement detector and $H_{a}$ denotes the transfer function of electrostatic force actuator.

**Figure 6 sensors-22-04930-f006:**
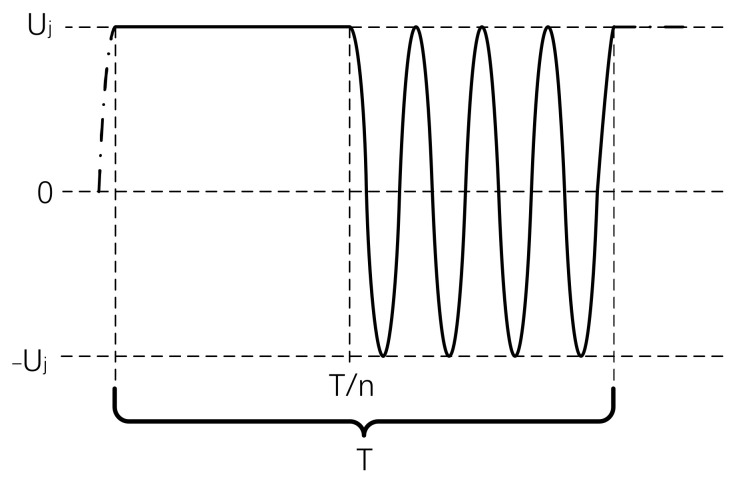
The waveform of the applied voltage, pattern in the solid line indicates a single period of the signal.

**Figure 7 sensors-22-04930-f007:**
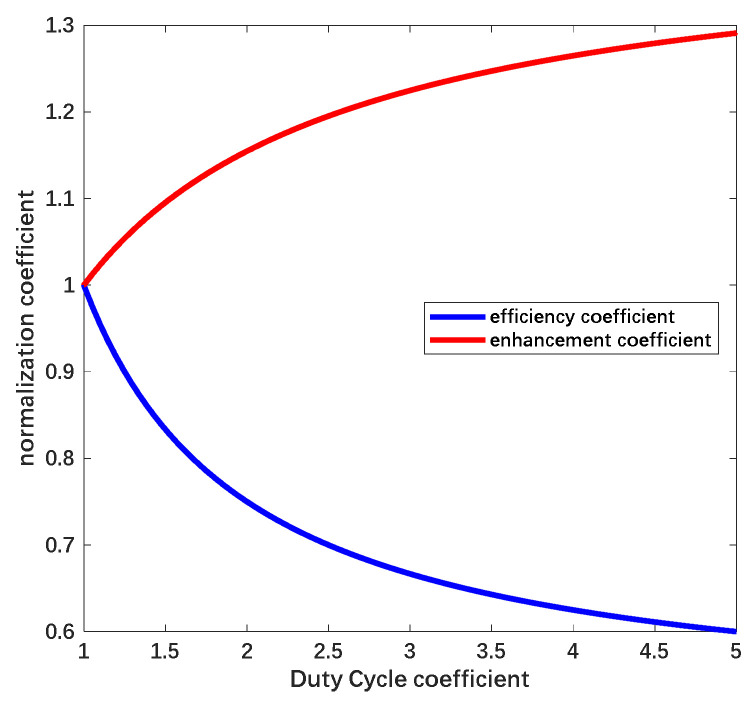
Relationship between the duty cycle of the signal and the coefficient of efficient force and magnification.

**Figure 8 sensors-22-04930-f008:**
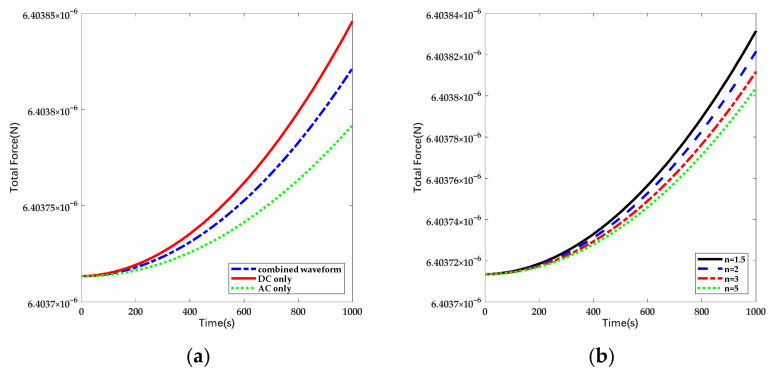
The variation of total electrostatic force with linearly increasing deposited charge in time domain: (**a**) Total electrostatic force under different forms of actuation and the coefficient of duty cycle is set as 2; (**b**) Total electrostatic force under different partition of duty cycle.

**Figure 9 sensors-22-04930-f009:**
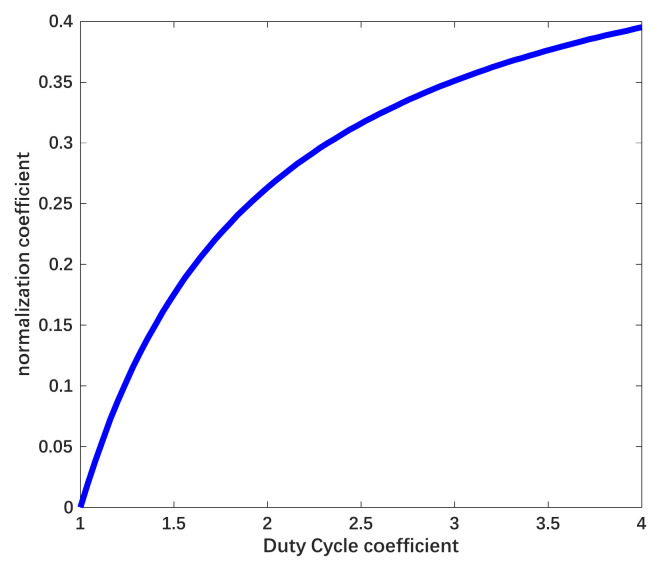
Relationship between the duty cycle and the coefficient of disturbance suppression.

**Figure 10 sensors-22-04930-f010:**
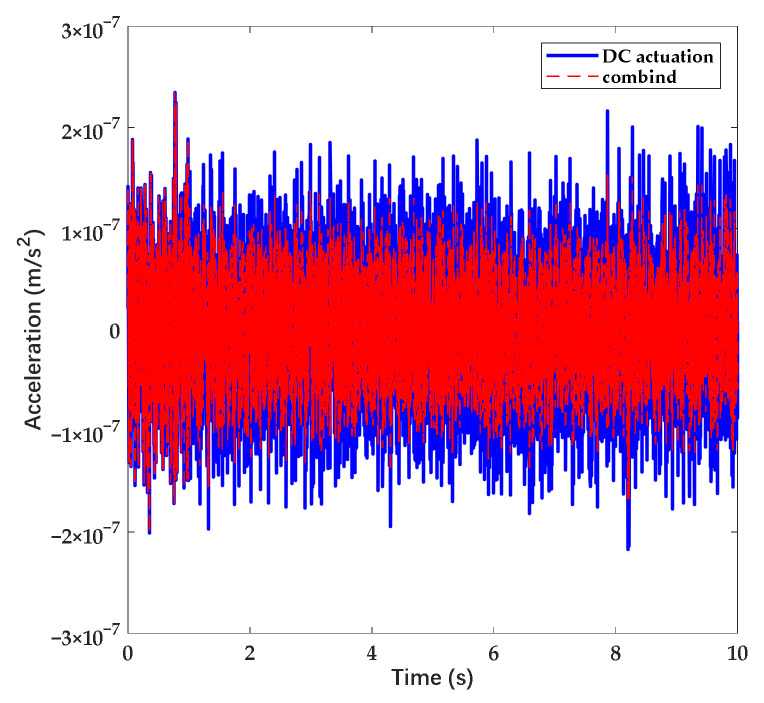
Acceleration error while applying DC actuation and combined actuation. The coefficient n increases with time.

**Figure 11 sensors-22-04930-f011:**
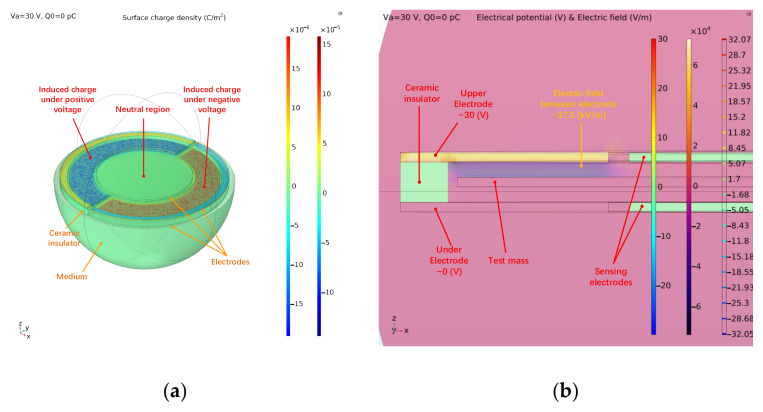
Simulation model for simplified electrode pair: (**a**) Distribution of surface charge density on the mass; the fixed electrodes are set as transparent; (**b**) Distribution of space potential and electric field between electrodes.

**Figure 12 sensors-22-04930-f012:**
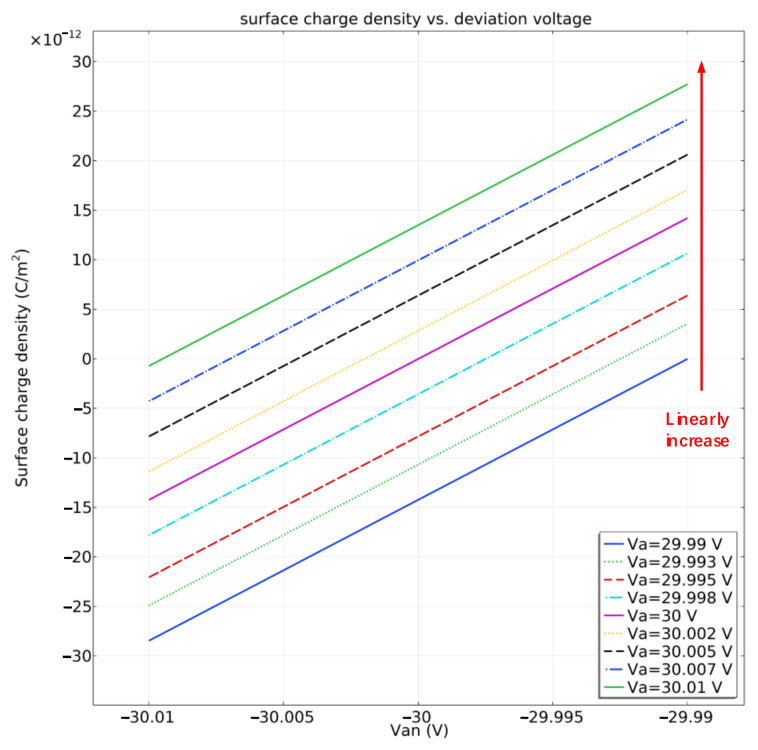
Variation of surface charge density of neutral region with fluctuating applied voltage.

**Figure 13 sensors-22-04930-f013:**
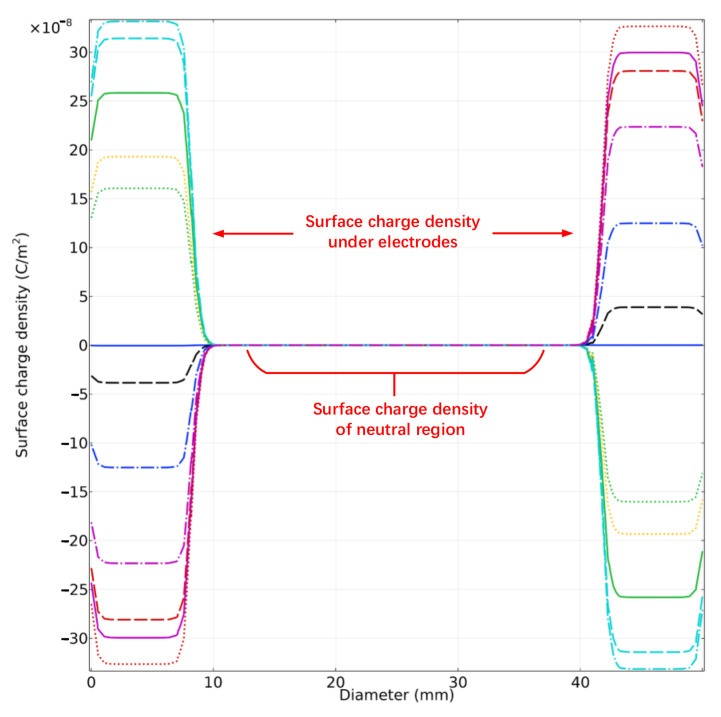
Distribution of surface charge density on whole surface under AC actuation. The curve indicates the distribution at discrete time points.

**Figure 14 sensors-22-04930-f014:**
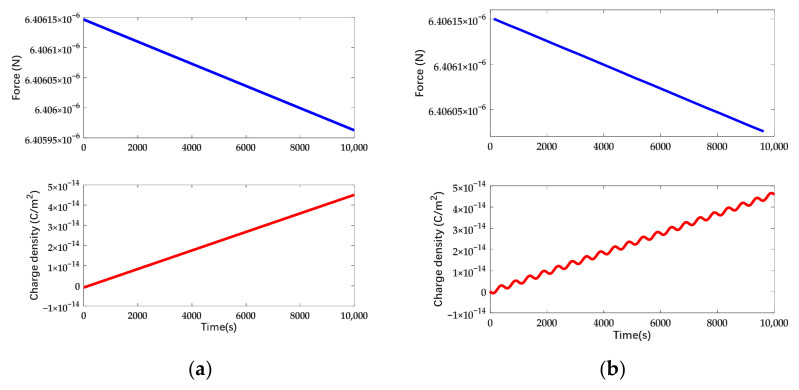
Charge density and electrostatic force variation in time domain while the deposited charge is increasing. The actuation voltage is symmetrically applied: (**a**) Applying DC actuation; (**b**) Applying AC actuation.

**Figure 15 sensors-22-04930-f015:**
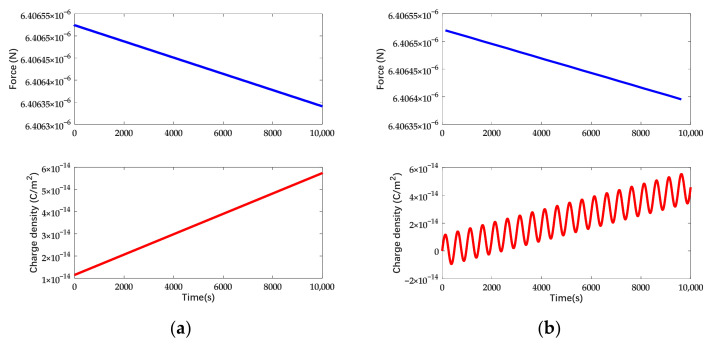
Charge density and electrostatic force variation in time domain while the deposited charge is increasing. The actuation voltage is asymmetrical and the deviation voltage is 1 mV: (**a**) Applying DC actuation; (**b**) Applying AC actuation.

**Figure 16 sensors-22-04930-f016:**
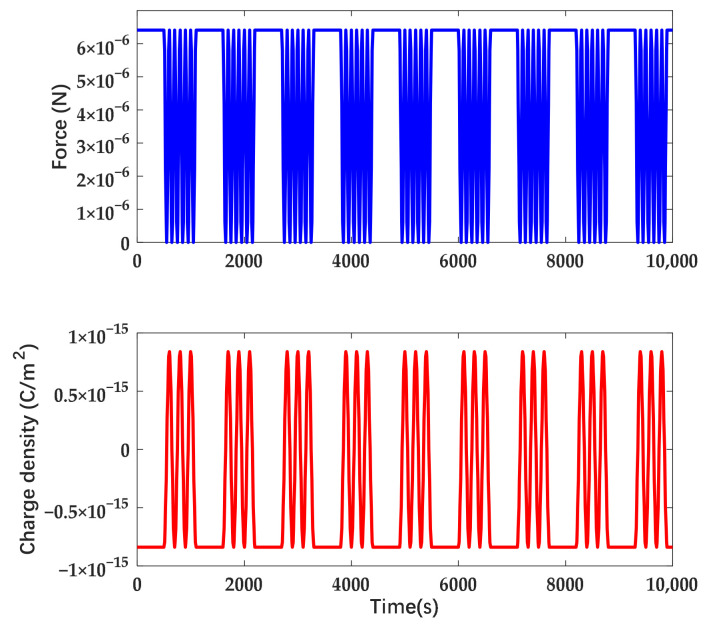
Electrostatic force and surface charge density in neutral area under actuation in the combined waveform.

**Figure 17 sensors-22-04930-f017:**
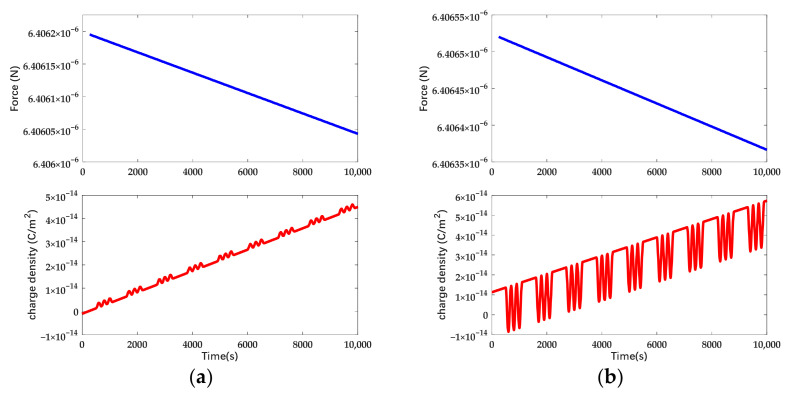
Charge density and electrostatic force variation under actuation in combined waveform while the deposited charge is increasing: (**a**) The actuation voltage is symmetrically applied. (**b**) The actuation voltage is asymmetrical and the deviation voltage is 1 mV.

**Figure 18 sensors-22-04930-f018:**
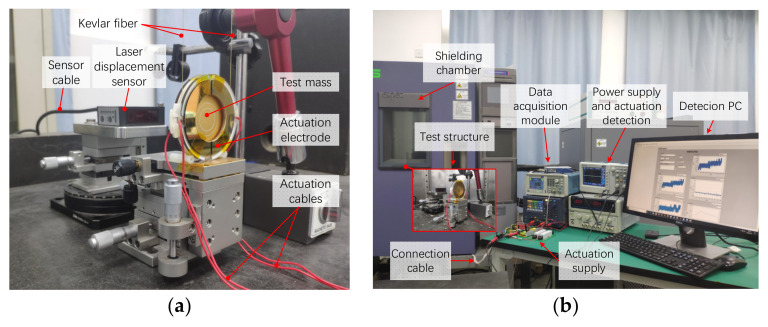
Testing structure for electrostatic force measurement: (**a**) The suspended pendulum and optical sensor fixed on motivation platform. (**b**) Set-up of force measurement experiment.

**Figure 19 sensors-22-04930-f019:**
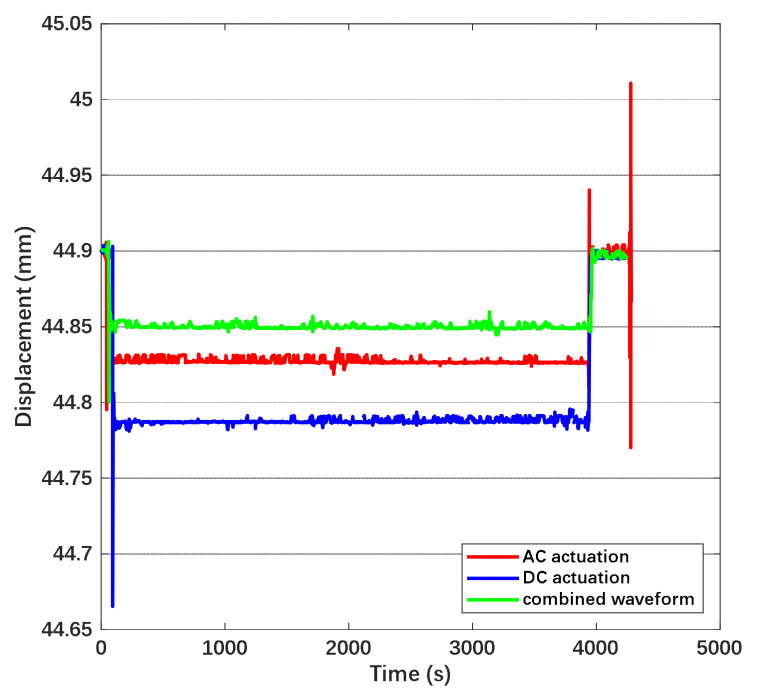
Displacement of the wafer in time domain. The amplitude of DC actuation and the combined waveform is 80 V and the amplitude of AC actuation is 143 V. The coefficient n is set as 1.67.

**Figure 20 sensors-22-04930-f020:**
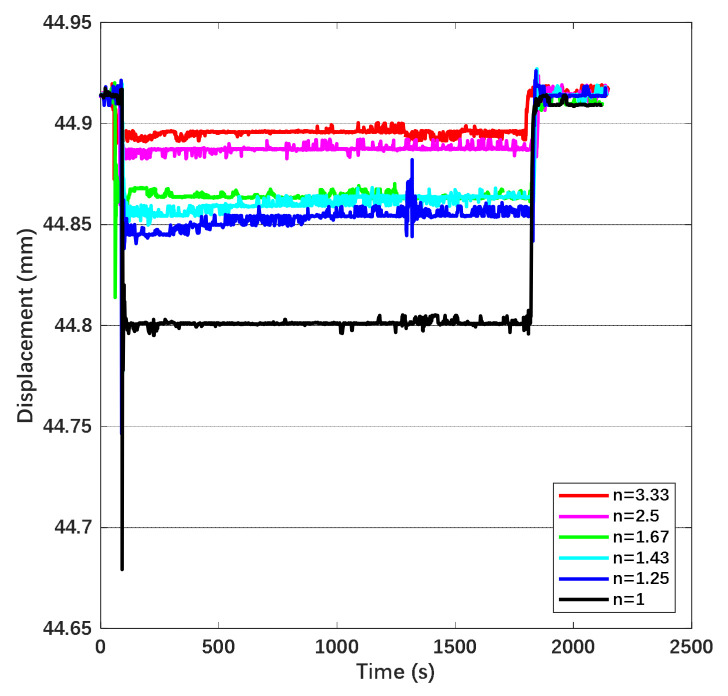
Displacement variation in time domain of different coefficient n with the same amplitude.

**Figure 21 sensors-22-04930-f021:**
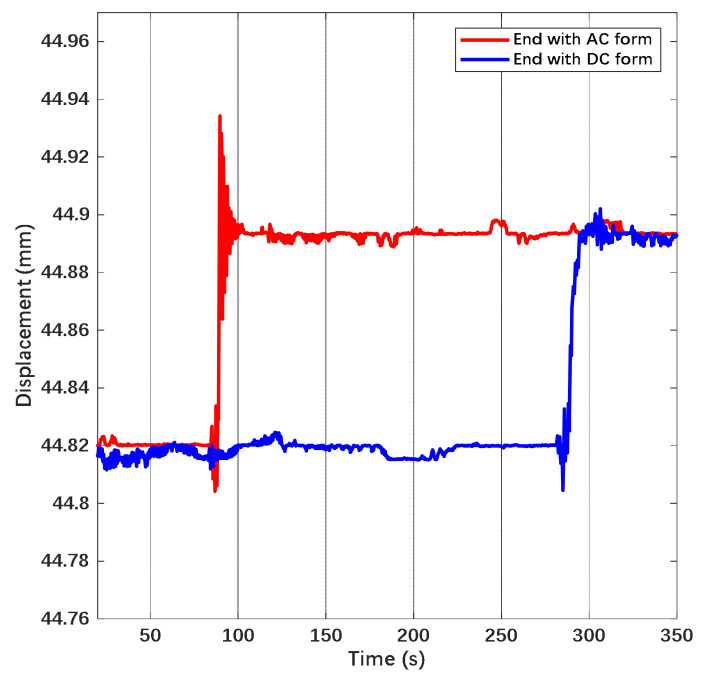
Displacement variation in time domain after switching off the source.

**Table 1 sensors-22-04930-t001:** Relevant ESA parameters used for evaluation.

Name	Value
Actuation capacitance	11.384 pF
Total capacitance	40.56 pF
Distance	0.8 mm
DC Actuation voltage	30 V
Amplitude of AC signal	42.4 V
Actuation frequency	500 Hz
Voltage deviation	1 mV

**Table 2 sensors-22-04930-t002:** Suppression efficiency under different forms of actuation.

Form of Actuation	Deviation Voltage (V)	Suppression Efficiency
DC actuation	0/0.001/0.05	0%
AC actuation	0	32%
AC actuation	0.001	33.5%
AC actuation	0.05	33.6%
Combined waveform (n = 2)	0	17.8%
Combined waveform (n = 2)	0.001	21.1%
Combined waveform (n = 2)	0.05	21.3%
Combined waveform (n = 3)	0.05	23.9%
Combined waveform (n = 1.5)	0.05	14.6%

**Table 3 sensors-22-04930-t003:** Variation of deposited charge and displacement drift under different coefficient.

Coefficient	Time Drift	Equivalent Deposited Charge	Suppression Efficiency
1	0.00455 mm/h	13.015 Pc	0%
1.25	0.0042 mm/h	12.504 Pc	3.93 %
1.43	0.004 mm/h	12.203 Pc	6.24 %
1.67	0.003 mm/h	10.568 Pc	18.8%
2.5	0.0021 mm/h	8.842 Pc	32.1%
3.33	0.0015 mm/h	7.473 Pc	42.6%

## Data Availability

Not applicable.

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
