# Peer review of "Study on the Method of Charge Accumulation Suppression of Electrostatic Suspended Accelerometer"

_sensors, 2022, doi:10.3390/s22134930_

Round 1
Reviewer 1 Report
Line 24 and 25: Please remove these statements “The introduction should briefly place the study in a broad context and highlight why it is important. It should define the purpose of the work and its significance”.
Line 74: please mention briefly the composition of the paper at the end of the introduction section; for example, section 2 discussed the theory while section 3 illustrated …………..
Line 154: please refer to figure 2 in the context: "as shown in figure 2"
Line 252: According to the figure; Which figure? Please specify which figure you are referring to?
Figure 10: Could you make this figure clearer?
Figure 11: Same as the previous comment and please enhance the resolution of this figure
Figure 12: Same as the previous comment
Figure 9: please put the units for x label and y label in brackets ()
for example, Time (s)
Figures 13, 14, and 15: Same as the previous comment
Author Response
Dear Reviewer:
On behalf of my co-authors, we thank you very much for your comments concering our manuscript entitled " Study on the method of charge accumulation suppression of electrostatic suspended accelerometer"(sensors-1725729). Those comments are all valuable and very helpful for revising and improving our manuscript, as well as the important guiding significance to our researches. We have studied the comments carefully and have made revision which we hope meet with approval. The responds to the comments are listed in the attachment.
Once again, thank you very much for your comments and suggestions.

Reviewer 2 Report
The charge accumulation is a big problem for the current electrostatic suspended accelerometer. In this manuscript, the authors proposed the application of actuation voltage with a combined waveform to suppress the acceleration noise due to deposited charge. The disturbance due to the charge can be suppressed by up to 40%, which is very impressive. For acceptance, the authors should answer the following questions.
1. In the introduction, it will be better if the sentences “The introduction should briefly place the study in a broad context and highlight why it is important. It should define the purpose of the work and its significance.” can be deleted.
2. In the introduction, it will be better if some references to other kinds of the accelerometer can be cited
(1) Capacitive: Mukhiya, R., et al. "Design, modelling and system level simulations of DRIE-based MEMS differential capacitive accelerometer." Microsystem technologies 25.9 (2019): 3521-3532.
(2) Piezoresistive: Liu, Feng, et al. "Optimal design of high-g MEMS piezoresistive accelerometer based on Timoshenko beam theory." Microsystem Technologies 24.2 (2018): 855-867.
(3) Piezoelectric: Gesing, A. L., et al. "On the design of a MEMS piezoelectric accelerometer coupled to the middle ear as an implantable sensor for hearing devices." Scientific reports 8.1 (2018): 1-10.
(4) Resonant: Ding, Hong, Changju Wu, and Jin Xie. "A MEMS resonant accelerometer with high relative sensitivity based on sensing scheme of electrostatically induced stiffness perturbation." Journal of Microelectromechanical Systems 30.1 (2020): 32-41.
3. It will be better if the structure parameters of the device are listed in a table combined with the structure scheme of the device, the configuration of AC and DC should also be indicated in a figure. Please put this figure before Figure 1.
4. Figure 10, 11, and 12 have too low resolution, please solve this problem. Some essential notes should be added to Figure 10 and 17.
5. Why the simulation doesn’t show the 40% charge suppression?
Author Response

(The authors gave the same response as above.)

Reviewer 3 Report
In this paper, the authors present a method to suppress charge noise in an electrostatic accelerometer. The argument is interesting, but the article shows some issues that need to be fixed:
1) The authors must discuss the novelty and the advantages of the proposed method in a more detailed way. In particular, a comparison among the performances of the proposed method and the other approaches proposed in the literature has to be carried out, mainly referring to the field of application of the considered device.
2) The authors must clarify the relationship between the numerical model analyzed by using Comsol multiphysics (see fig. 7) and the so-called “testing structure” (see fig.17) employed for realizing the electrostatic force measurements. Furthermore, the authors must explain how the theoretical results and measurements are related. i.e., the theoretical indications were useful to carry out the campaign of measurements.
3) What is the formal definition of “suppression efficiency” used by authors in their study?
4) The authors must provide pictures of i) the actual testing structure, ii) the actual instrumentation equipment used in their experimental campaign.
Furthermore, some typos need to be fixed, i.e.
1) Page 1, lines 24-25: the statements “The introduction should briefly place the study in a broad context and highlight why it is important. It should define the purpose of the work and its significance.” should be deleted.
2) Page 6, lines 146-147: the statement “substituting equations 11,12 to 8, the electrostatic force with asymmetric voltage distribution can be attained as” should be fixed (since the equation that follows this statement is precisely the equation (12))
Author Response

(The authors gave the same response as above.)

Round 2
Reviewer 3 Report
The authors have answered all my concerns. The paper can be accepted for publication.